# Extracellular Vesicles as Promising Carriers in Drug Delivery: Considerations from a Cell Biologist’s Perspective

**DOI:** 10.3390/biology10050376

**Published:** 2021-04-27

**Authors:** Giona Pedrioli, Ester Piovesana, Elena Vacchi, Carolina Balbi

**Affiliations:** 1Laboratory for Biomedical Neurosciences, Neurocenter of Southern Switzerland, Ente Ospedaliero Cantonale, 6807 Taverne-Torricella, Switzerland; giona.pedrioli@outlook.com (G.P.); ester.piovesana@eoc.ch (E.P.); elena.vacchi@eoc.ch (E.V.); 2Faculty of Biomedical Sciences, Università della Svizzera Italiana, 6900 Lugano, Switzerland; 3Laboratory of Cellular and Molecular Cardiology, Istituto Cardiocentro Ticino, 6807 Taverne-Torricella, Switzerland; 4Center for Molecular Cardiology, University of Zurich, 8952 Schlieren, Zürich, Switzerland

**Keywords:** extracellular vesicles, exosomes, apoptotic bodies, microvesicles, drug delivery, therapeutic potential, precision medicine, cell-free therapy

## Abstract

**Simple Summary:**

Extracellular vesicles are promising nanocarriers of active pharmaceutical ingredients in precision medicine. However, to develop safe and effective extracellular vesicles-based therapies, it is crucial to gain fundamental knowledge on key cell biology mechanisms governing the extracellular vesicles-mediated cell-cell exchange of macromolecules. In this review, we present five aspects we believe are worth taking into account before considering extracellular vesicles as candidate vectors of pharmaceutical ingredients in cell-free therapies.

**Abstract:**

The use of extracellular vesicles as cell-free therapy is a promising approach currently investigated in several disease models. The intrinsic capacity of extracellular vesicles to encapsulate macromolecules within their lipid bilayer membrane-bound lumen is a characteristic exploited in drug delivery to transport active pharmaceutical ingredients. Besides their role as biological nanocarriers, extracellular vesicles have a specific tropism towards target cells, which is a key aspect in precision medicine. However, the little knowledge of the mechanisms governing the release of a cargo macromolecule in recipient cells and the Good Manufacturing Practice (GMP) grade scale-up manufacturing of extracellular vesicles are currently slowing their application as drug delivery nanocarriers. In this review, we summarize, from a cell biologist’s perspective, the main evidence supporting the role of extracellular vesicles as promising carriers in drug delivery, and we report five key considerations that merit further investigation before translating Extracellular Vesicles (EVs) to clinical applications.

## 1. Introduction

Intercellular communication is essential for multicellular organisms’ functions and homeostasis. Cells have evolved a variety of mechanisms to exchange biomolecules and ensure the synchronization of cellular activities. In 1983, Johnstone and colleagues [1] discovered a novel mechanism used by cells to communicate, which involves the secretion of lipid-bilayer membrane nanovesicles (later named extracellular vesicles; EVs) in the extracellular space. EVs were initially considered as a way for cells to get rid of “cytoplasmic waste” such as membrane debris or protein aggregates [2], but in 1996, Raposo et al. [3] showed that EVs were able to stimulate the adaptive immune response. Since then, EVs have been extensively studied and now are recognized as vehicles for the cell-to-cell exchange of biomolecules that support critical cellular functions. EVs are classified into three major groups based on their biogenesis: apoptotic bodies (ApoBDs), microvesicles (MVs), and exosomes (EXO) [4] (Figure 1). ApoBDs are released by apoptotic cells and their size has been reported to range between 1 to 5 µm. The lumen of these vesicles contains fragments of cytoplasm and organelles belonging to their cells of origin. For instance, the most used markers for such EVs are two chaperon proteins of the endoplasmic reticulum (ER), calnexin, and endoplasmin (GRP94), likely resulting from the fragmentation of the ER during apoptosis [5]. MVs, their size ranges from 100 nm to 1 µm, are formed by the outward budding of the producing cell plasma membrane [6]. A detailed proteomic comparison between different types of vesicles identified the mitochondrial inner membrane protein (mitofillin) and the alpha-actinin-4 (actinin-4) as proteins enriched in the MVs population [7]. EXO are instead classified as the smallest vesicles. Their size ranges from 30 to 150 nm and slightly overlaps with some MVs. However, MVs and EXO can be distinguished on the basis of specific proteins that participate in their different mechanisms of biogenesis. Indeed, EXO’s biogenesis is strictly associated with the endocytic pathway and, in particular, with the endosomal multivesicular bodies’ (MVBs) formation. Two major biogenesis pathways of EXO are known; the first depends on the endosomal sorting complexes required for transport (ESCRT-dependent) and the second is known as ESCRT-independent mechanism. Both pathways are involved in (1) protein sequestration and modification, (2) processing and tracking of the resulting vesicles, and (3) their fusion to the plasma membrane [8]. ESCRT-dependent EXO are enriched in syntenin-1 and ALG-2-interacting protein X (ALIX), two proteins that participate in the formation of intraluminal vesicle (ILVs) and cargo sequestration [9]. ESCRT-independent EXO are instead enriched in tetraspanins, in particular CD9, CD63, CD81, that orchestrate the formation of the tetraspanin-enriched microdomains (TEMS) in the plasma membrane [10,11] and in ceramide, a simple sphingolipid (SL), which plays a critical role in membrane biogenesis [12]. Both pathways, ESCRT-dependent or -independent, work synergistically.

Although there are no specific markers that can uniquely discriminate against the three different classes of EVs, the continuous study of biogenesis and identification of specific protein involved are essential for improving the isolation and characterization of different subtypes of EVs.

The increasing knowledge of the different EV subtypes and the interest in the research on their roles has shown that these vesicles are implicated in health (e.g., tissue repair [13,14], stem cell maintenance [15], immune response [16,17]) and disease [18,19]. Furthermore, there is growing evidence that demonstrates the potential of EVs as carriers for efficient drug delivery [17,20,21]. In this review, we introduce, from a cell biology perspective, the latest findings supporting a role of EVs as therapeutic drug carriers. We highlight some important aspects that we believe are worth considering when designing and manufacturing EVs for drug delivery.

## 2. Therapeutic Potential of Stem Cell Derived-EVs

In the last years, stem cell-based therapies have been studied with the goal to address the need to replace and heal injured tissues. Stem cells were indeed considered as a key element of regenerative medicine therapies, due to their inherent ability to differentiate into a variety of cell types. While the beneficial effect obtained in different fields (e.g., cardiovascular medicine [22] or bone regeneration [23]) was largely validated, the initial mechanistic hypothesis of trans-differentiation has been lately questioned. To date, the beneficial effect reported were later mainly associated with their ability in the stimulation of the local microenvironment [24,25]). This has led to a paradigm shift, where the trophic molecules secreted by the transplanted cells are now considered more critical than the differentiation potential of the cells. Secreted EVs prominently figure among vectors of signals that regulate cell function. By transporting biologically active molecules from cell to cell, EVs are key mediators of intercellular communication. For the mentioned reasons, in the last decade, an increase number of studies indicate that EVs, predominantly derived from stem or progenitor cells, have a therapeutic potential (Figure 2). EVs can deliver immunomodulatory signals [26] (Table 1), stimulate cell proliferation [20], induce angiogenesis [27], suppress apoptosis [28], and drive tissue regeneration [29]. In regenerative medicine the application of EVs derived from multipotent stem cells has been highly investigated. EVs isolated from mesenchymal stem cells (MSCs) have shown beneficial effects in healing a variety of stressed tissues. He et al. showed that treatment with MSC-derived EVs (MSC-EVs) protect against kidney damage in a subtotal nephrectomy murine model of renal regeneration. Treatment with MSC-EVs decreased the levels of uric acid, creatinine, fibrosis, and lymphocyte infiltration [30]. Furthermore, MSC-EVs triggered macrophage polarization in vitro and in vivo in a cardiotoxin-induced murine model [17]. The therapeutic potential of MSCs-EVs was also demonstrated in different disease models such as a skin burn model [31] and skin defects [32], hindlimb ischemia [33], sepsis [34], and myocardial infarction [35,36,37]. EVs derived from fetal MSCs, such as human amniotic fluid derived stem cells (hAFS) were also investigated for their therapeutic capacity. A deep characterization of hAFS-secreted EVs (hAFS-EVs) demonstrate their ability to induce proliferation, survival, immunomodulation, and angiogenesis in different in vitro and in vivo models [20]. Furthermore, hAFS-EVs increased cardiac function, while reducing inflammation and scar formation, when intracardiac was injected into a mouse model of myocardial infarction [29]. The therapeutic effect of EVs in myocardial infarction was also exploited using EVs obtained from cardiac mesenchymal progenitor cells (CPCs) or cardiosphere derived cells (CDCs). In 2014, two independent studies showed the cardioprotective role of produced EVs after myocardial ischemia. Ibrahim A. et al. demonstrated that EXO are the active component of the CDC secretome in triggering cardiac repair [38]. In particular, miR-146a-5p was found to be enriched in vesicles and responsible for cardiomyocyte protection after myocardial infarction injury via targeting of IRAK-1 and TRAF6 [39], both involved in the toll-like receptor (TLR) signaling pathway [40]. In addition, miR-146a-5p suppresses NOX-4 [41], which has been shown to increase oxidative stress enhancing cardiac injury [42], and SMAD4, a member of the transforming growth factor beta (TGF-beta) profibrotic pathway [43]. Barile et al., at the same time, showed that EVs from CPCs are cardioprotective and protect cardiomyocytes from apoptosis while inducing new vessel formation [28]. Furthermore, a second study from the same group identified pregnancy-associated plasma protein-A (PAPP-A) as a protein carried by CPCs-EVs and responsible for their cardioprotective effect [44].

All these studies suggest that naïve EVs secreted by stem or progenitor cells have beneficial properties that may be exploited as future therapies.

It is important to underline that not only stem or progenitor cells are able to secrete EVs. As a matter of fact, almost all types of cells produce EVs, and their role in the organisms could be not only physiological but also pathological [45]. For example, EVs play an essential role in both primary tumor growth and metastatic evolution [46,47]. For the above reasons, it is important to study and understand, from a cell biology point of view, the role not only of EVs but also of their secreting cells, when used as therapeutic tools.

## 3. Can EVs Act as Drug Delivery Carriers?

Thanks to their biological proprieties EVs represent a promising carrier for drug delivery. Indeed, using EVs as “drug carriers” provides several advantages, such us:-low immunological response [48];-targeting potential [49];-easy bypass of cellular barriers [50];-cargo protection [51].

Besides their natural biocompatible characteristics [52], EVs have a low immunological response. In particular, if derived from an appropriate cell source, such as MSCs, CPCs, or Dendritic Cells (DCs), there is no risk of rejection. Furthermore, EVs can be directly derived from the patients [53].

Another important property of EVs is their target potential. Their asymmetrical lipid distribution and specific protein composition of the membrane justify their organotropism and homing ability [54]. Interestingly, Qiao and colleagues recently demonstrated that tumor cell-derived exosomes are preferentially home to their cells of origin [55]. The biocompatibility of EVs is also reflected in their ability to bypass cellular barriers. One of the most interesting barriers is the Brain Blood Barrier (BBB). Alvarez-Erviti et al. demonstrated the efficacy in vivo of EVs to surmount the BBB and deliver shRNA to target cells [56]. Despite different studies that reported this ability, the mechanism of action still needs to be determined. In an attempt to clarify how EVs overcome BBB, the in vitro interaction between EVs and brain microvascular endothelial cells (BMECs) was explored [57]. Chen and colleagues found that only in inflammatory conditions, EVs were able to cross the BMECs monolayer and that the mechanism is transcellular: EVs were uptaken by endocytosis followed by MVB formation and then exocytosis.

Another important advantage of EVs-based therapy is represented by their capacity in cargo protection. Indeed, while the delivery of exogenous RNA into the bloodstream leads to its rapid degradation, the lipid bilayer of EVs acts as a protection barrier against RNAses [58].

For the discussed benefit, an increased number of laboratories have focused their research on EVs as a carrier system for drug delivery. In 2015, the International Society for Extracellular Vesicles (ISEV) came out with a position paper revising the role of EVs in the clinical trial [59]. Here, they presented several translational studies on EVs, highlighting how, so far, no recommended standard techniques have been established for the clinical grade production and quality control of EV-based therapeutics. Furthermore, several considerations in manufacturing and safety must be implemented. In particular, even if EV-based therapeutics are defined as biological drugs (for which the European, Australian, and United States’ regulatory frameworks for manufacturing and clinical trials exist) special guidelines targeting EV-based therapeutics may be needed.

It is clear that there are still multiple aspects and open questions on EVs in drug delivery that range from difficulties in isolation and purification to low cargo transfer in target cells [60,61]. In the next five chapters, we report five aspects that we believe are worth considering to address when moving to EV based applications (Figure 3).

### 3.1. First Aspect: EV Isolation and Purification

EVs are mainly isolated from natural sources such as the conditioned medium of cultured cells or biological fluids. Cells secrete a high heterogeneity of vesicles whose composition and physicochemical properties are dependent on parental cell type, cellular activation state, local microenvironment, biogenesis, and cargo sorting [62] (Table 2). In biological drug manufacturing, a critical step is to purify the therapeutic drug away from other biological components [54]. Scalable isolation methods that yield high purity of the preparation are essential for industrial manufacturing. EVs are purified through conventional isolation methods that exploit biochemical and biophysical characteristics of lipid bilayer vesicles and protein complexes [60,62]. For instance, immune affinity strategies purify EVs binding molecules that are exposed on their surface. Density gradient ultracentrifugation (DG-UC) is the gold standard method to purify EVs on the basis of their density, while size-exclusion chromatography (SEC) and tangential flow filtration (TFF) are used to isolate EVs on the basis of their size (Figure 3) [63]. Nevertheless, the high heterogeneity of EVs brings complexity in the isolation of pure EV subtypes. Recent guidelines issued by the leading society in the field (i.e., ISEV) suggest combining multiple techniques that purify EVs on the basis of different biochemical and biophysical properties [64]. For example, the sequential use of SEC and DG-UC isolates small EVs from biofluids with higher confidence [65]. However, a combinatory approach is not suitable for scale-up production, as the purification results in a low yield of EVs [66]. Furthermore, it remains challenging to obtain pure preparations of EV subtypes that share similar properties [63]. The controlled condition to produce synthetic vesicles replicating EVs secreted from living cells may overcome challenges encountered in EV purification [67]. However, their potential in mimicking the therapeutic effects of EVs remains to be investigated.

The EV field is working hard to implement standardized purification methods to isolate pure EV subtypes. However, will these methods be suitable for scale-up production without compromising EV purity and/or integrity?

### 3.2. Second Aspect: EV Production Scale-up and Storage

As mentioned above, another important aspect to be considered in the process of translation of EVs from the research contest to pharmaceutical use is the scale-up production. Scale-up for cells could result in the induction of apoptotic blebs, downregulation of interleukin receptors, induction of mechanically stimulated protein-kinase pathways, and modulation of extracellular signal-related kinase [68] (Table 2). There are reports of the reduced proliferative capacity of primary cells in scaling up, with the consequent limitation of the batch production number or reproducibility [68]. For such reasons, the potential change in the cellular phenotype during technical transfer (e.g., scale-up) must be considered. Another issue is that the scaling-up of adherent cell cultures requires technologies that maximize the culture surface area, such as microcarriers in stirred-tank reactors or culture cell-tower flasks [69].

Industrial development requires high-efficacy manufacture and translatable isolation techniques compliant with good manufacturing practices (GMPs). Several GMP-grade manufacturing methods for large-scale EVs production, based on different technologies, have been recently made available [21,70]. Looking at isolation, TFF is a closed system that can be easily sanitized in compliance with GMPs and actually represent the most used technique to isolate EVs in a scale-up production.

Finally, the storage of produced EVs is another additional factor that can impact the amount and quality of the final product. The use of siliconized vessels in the purification and storage prevent adherence to the surfaces and consequent loss of EVs [71]. Storage at −80 °C in phosphate saline buffer (PBS) is currently the most commonly adopted method, since different studies addressed that 4 °C storage causes EV damage and aggregation [72,73]. However, freeze/thaw cycles should be minimized, as they may damage the EV membranes. In this regard, adding cryoprotectants, such as trehalose, appears to have a positive impact on EVs [74]. The already well known use of trehalose to stabilize labile proteins [75] and liposomes [76] is a step forward towards a quick determination of a standard GMP-grade storage protocol (Figure 3).

### 3.3. Third Aspect: EV Cargo Loading

The efficient packaging of active pharmaceutical ingredients (API) into biological carriers is a critical step in drug manufacturing [77] (Table 2). Biologically active macromolecules transported by EVs, best known as cargo, are the APIs that ensure a biological effect in target cells. The lipid-bilayer membranes of EVs act as protective shields for soluble APIs transported within their lumen [78]. Cells have evolved specific mechanisms for the loading of macromolecules within EVs. For instance, ubiquitinated proteins are brought to the lumen of intraluminal vesicles, the intracellular equivalence of EXO, by the activity of ESCRT machinery [79]. Similarly, the enrichment of specific miRNAs within EVs involves the participation of a protein accessory of the ESCRT machinery, i.e., ALIX, in association with the protein argonaute-2 (AGO2) [80]. On the other hand, membrane-bound cargos are likely to require the participation of different pathways. For example, membrane-bound tetraspanins inserted into the plasma membrane by the secretory pathway are routed back to endosomal compartments, take part in the generation of EXO, and become constituents of their membranes [81]. Approaches that exploit these endogenous machineries to package APIs into EVs are widely used in R&D [82]. Most of the time, simple overexpression, in the parental cell, of the desired APIs is sufficient to load EVs. For example, in 2010, Zhang and colleagues [83] showed, for the first time, that THP-1 monocytes transfected with miR-150 were able to secrete EVs enriched in miR-150 that was functionally delivered in recipient cells. In the last years, different studies have confirmed this RNA transfer model, not only with miRNA but also with shRNA [84,85]. Recently the same approach of overexpression of the angiotensin-converting enzyme 2 (ACE2) receptor in parental cells was used to generate EVs enriched in ACE2 (EVs-ACE2). Interestingly Cocozza et al. showed that in vitro co-treatment with EVs-ACE2 and SARS-CoV2 pseudovirus significantly reduce the infection of target cells [86]. The same strategy was used in a parallel work [87].

In other instances, APIs are poorly sorted into EVs, thus requiring molecular facilitators for their successful enrichment [88,89]. A strategy exploited is the tagging of proteins with a peptide that has a high affinity for ubiquitination; this results in an increased enrichment into EVs [88]. Incorporation of interest protein in a DNA vector fused with Nef exosome-anchoring also results in an improvement in the specificity and loading efficiency of targeted proteins in exosomes [90].

In biomanufacturing, the artificial incorporation of APIs into EVs can be achieved post-EVs isolation (Figure 3). The most straightforward adopted solution is passive cargo loading. This method involves the co-incubation of EVs and APIs in controlled conditions [91]. Hydrophilic APIs tend to diffuse towards lipid-membranes and localize in the EVs lumen following a concentration gradient [92].

To enhance EVs cargo loading, a strategy is to coat the drug with synthetic nanoparticles smaller in size (5–30 nm) such as silica or carbons nanoparticles (CNPs) [93]. Illes and colleagues recently exploited the role of hybrid nanoparticles of the metal−organic framework (MOF) as a valuable alternative for drug encapsulation into EVs [94].

Furthermore, to improve the loading efficiency, EVs can be physically or chemically manipulated. Electroporation, sonication, and cycles of freeze-thaw, to name a few, are active cargo loading mechanisms that temporarily disrupt the lipid membranes of EVs and increase the efficiency of the packaging [95,96,97,98].

Other loading methods rely on the surface charge. EVs, being overall negatively charged, can undergo electric and magnetic fields to induce cargo insertion or release [99,100]. It is important to note that all these techniques can increase loading efficiency but, on the other hand, are often associated with negative effects such as membrane integrity loss, aggregates formation and cargo impurity.

While several techniques are available to enrich APIs into EVs efficiently, technological advances will have to demonstrate controlled and flexible loading of the APIs for personalized applications.

### 3.4. Fourth Aspect: EV Targeting

Targeting drugs to specific cells is the key in enhancing drug efficacy. Strategies aiming at increasing targeted drug delivery have several advantages over strategies that expose the organism systemically. In particular, targeted delivery reduces systemic drug exposure and the toxicity derived from off-target effects. EVs are candidate drug delivery vectors for targeted strategies thanks to their selectivity towards specific cell types. It is known that the trophism to specific organs could be influenced, at least to some extent, by EVs lipid composition and protein content. For example, the uptake of EVs from macrophage is associated with the presence of phosphatidylserine on the EVs surface [101] (Table 2) and the accumulation in brain, lungs, or liver with different types of integrin expressions on the EVs’ surface [102]. Tetraspanins, transmembrane proteins highly enriched on the surface of EVs, also participate in the EVs and cell recognition process. Dendritic cells selectively recognize and internalize EVs displaying CD9 and CD81 on their lipid-shield through a ligand-receptor mechanism [103]. In the context of tumors, EVs can be used to naturally target specifically cancer cells. EXO and small EVs have indeed the tendency to accumulate more into solid tumors than normal tissue; this phenomenon is mainly called Enhanced Permeability and Retention (ERP) [104]. Profit by this effect, injection of, for example, doxorubicin loaded EXO, could be used to selectively target solid tumors decreasing anticancer drugs side effects [105]. Artificial integration of ligands in EVs membranes is a common strategy used to enhance the selective tropism to target cells. Alvarez-Erviti and colleagues fused the rabies viral glycoprotein (RVG) to the amino terminus of the lysosome-associated membrane protein 2B (LAMP2B), a protein displayed on the surface of EVs, and augmented the targeting to cells expressing the acetylcholine receptor [56]. RVG is indeed a known protein that facilitates the fusion of viruses to the membranes of cells exhibiting the acetylcholine receptor [106]. LAMP2B is one of the most used strategies to increase EV targeting (Figure 3). Fusion of LAMP2B with a cardiomyocyte specific peptide (CMP) or with an iRGD peptide (a specific cycling peptide, able to target to alpha-V-beta-3 integrin, a protein mainly expressed in angiogenic vasculature tumor tissue, identified in an in vivo screening of phage display libraries in tumor-bearing mice [107]) was shown to efficiently target EVs to cardiomyocytes [108] or tumor tissue [109], respectively. Despite interesting results obtained with this targeting technique, some concerns have been raised regarding the efficiency of a peptide displaying on the vesicle surface [109]. The protection of proteolytic degradation can be achieved including glycosylation motifs (GNSTM) in the peptide-LAMP2B fusion [109]. Alternatively, to overcome the degradation problem, targeting peptides can be directly tethered to the plasma membrane. In this study, Ciullo et al. overexpressed the CXCR4 receptor on CPCs [110], resulting in EVs carrying CXCR4 (EVs-CXCR4). Intravenous injection of EVs-CXCR4 in a rat model of myocardial infarction increased cardiac uptake thanks to the formed SDF-1alpha gradient by the injured heart.

To sum up, harnessing EVs for targeted drug delivery has the potential to increase therapeutic efficacy as well as reduce side effects. However, the effectiveness of EV targeting strategies remains to be clarified. How can we protect EV surface ligands from proteolytic degradation? More in general, can EVs keep up their targeting promises for in vivo applications?

### 3.5. Fifth Aspect: EV Cargo Delivery

Reaching the desired target cells is not enough. To trigger a phenotypic, change to recipient cell-specific EV cargo delivery mechanisms are needed (Figure 3). The engagement between EV ligands and their cellular substrates is an obligatory step in triggering a cascade of reactions that alter cellular phenotypes. In principle, cargos transported on the surface of EVs can participate in a target engagement on the limiting membrane of cells. For instance, the binding of EV surface ligands to cell-membrane receptors has been described as a mechanism for T-lymphocytes mediated immune response via B-lymphocytes-derived EVs [3] (Table 2). In contrast, EV-soluble ligands are likely to need the delivery within the cytosol before engaging with their targets. The cytosolic environment is indeed rich in biomolecules originating in various subcellular compartments and having disparate cellular functions that could act as the molecular partner of the EV-soluble carried molecules. The fusion of EVs and recipient cell membranes has been described as a simple mechanism for the liberation of the EV content [111,112]. This event occurs at the limiting plasma membrane or within organelles of the endocytic pathway, reminiscing of mechanisms exploited by a virus to enter the cell [113]. Recent observation in cell culture conditions have brought new insights to EV-mediated target engagement that may not require the intermediate passage via cytosol, among which are the direct release of EV cargo within the nucleus and the encounter of endogenous and EV-transported proteins in acidic organelles of the cell [114,115].

At present, we do not completely understand the disparate mechanisms used by EVs to deliver their cargo to recipient cells. Can EV dictate phenotypic changes to target cells by simply acting at their surface? How does soluble EV cargo engage in a biologically relevant target engagement in recipient cells? The plethora of EV-transported biomolecules brings even more complexity to the system. Do EV-transported biomolecules share the same delivery mechanisms?

## 4. Conclusions

The number of clinical trials registered has more than doubled in the last three years. In January 2018, 66 clinical trials containing the words “exosome OR extracellular vesicles” were listed on the clinical trial database registry of the National Institute of Health [116]. Three years later, in March 2021, searching from the same entries resulted in 258 studies. Interestingly, only 45 studies resulted in being completed, and of these, only three are strictly related to the role of EVs as therapeutic drugs (NCT04491240; NCT01159288; NCT04276987), confirming that clinical translation is still at the beginning.

Although EVs are promising drug delivery carriers, several technical challenges have yet to be overcome before seeing EV-based therapeutics.

The first issue is represented by the choice of the appropriate EV isolation and purification methods. The EV field is working to standardize isolation and purification techniques to recover pure EV subtypes. While technological advances and standardized guidelines will improve EV purification, will these be suitable for scale-up and GMPs production? Alternatively, are we ready to accept a compromise between the requirement of GMP production and EV purity?

Looking more in detail at the scale-up and GMPs production, other concerns needs to be addressed. Producing cells need to be adequately studied and compared to the standard cultured one; indeed, potential changes in cellular phenotype during technical transfer (e.g., scale-up) are possible. Storage is also something that is still not standardized in the EV field, and further studies are needed.

Several questions remain unanswered regarding EV cargo loading, EV targeting, and EV cargo release. Even though several techniques are currently available to load APIs into EVs, standardized protocols that allow controlled and flexible loading have not yet been developed. Similarly, technological advances that will increase the EV specificity towards target cells/organs have the potential to increase effectiveness of EV-based therapies.

Lastly, we raised the issue of efficient cargo release that is currently influencing the use of EVs as therapeutic carriers. The main challenge here is represented by the lack of knowledge in the mechanism used by EVs to deliver their cargo in recipient cells. The different EV subtypes that may function as carriers of biomolecules add more complexity to the system. For such reasons, a lot of effort still needs to be dedicated for a more comprehensive knowledge of the EV cargo delivery.

All these points reflect the status of a low number of clinical trials that aim to use EVs as drug carriers. To note, the coronavirus pandemic situation caused by SARS-CoV2 spreading, has drastically increased the number of clinical trials using EVs. Indeed, several trials are currently ongoing that aim to use EVs as therapeutic agents against COVID-19 infection. In particular, an open-label Phase I clinical trial (NCT04747574) evaluates the safety of CD24-EXO in patients with moderate/severe COVID-19 disease. EVs are obtained from cells engineered to overexpress CD24. Isolated EVs are then aerosolized in normal saline for inhalation via a standard hospital-grade inhalation device. It is possible that, taking in account the critical worldwide situation caused by the pandemic, potential encouraging results obtained by modified inhalation of EVs will speed up even more the scientific community’s finding of the answers to the questions raised in this review, resulting in a standardized and efficient use of EVs as a drug delivery system.

## Figures and Tables

**Figure 1 biology-10-00376-f001:**
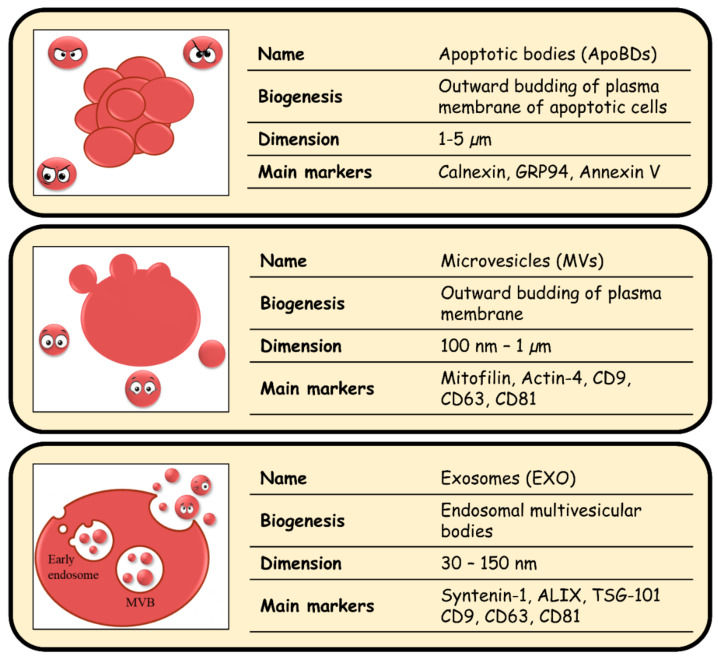
**Extracellular Vesicles** (EVs)-ID-CARD. *Schematic representation of the three main classes of EVs. For each group, biogenesis, dimension, and markers are reported*.

**Figure 2 biology-10-00376-f002:**
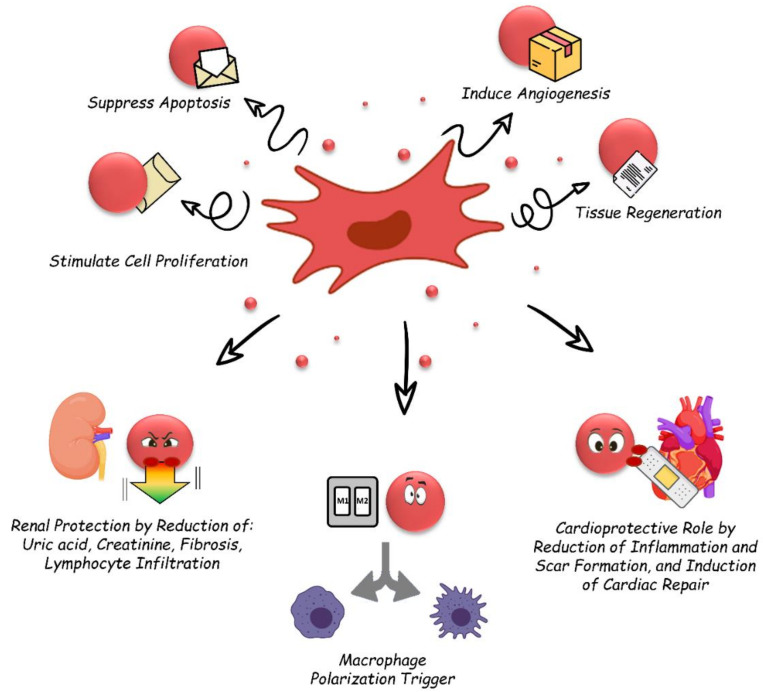
Therapeutic potential of stem cell derived-EVs. *Schematic representation of the suggested effects of EVs in regenerative medicine, with particular attention on renal and cardio protection, and macrophages polarization*.

**Figure 3 biology-10-00376-f003:**
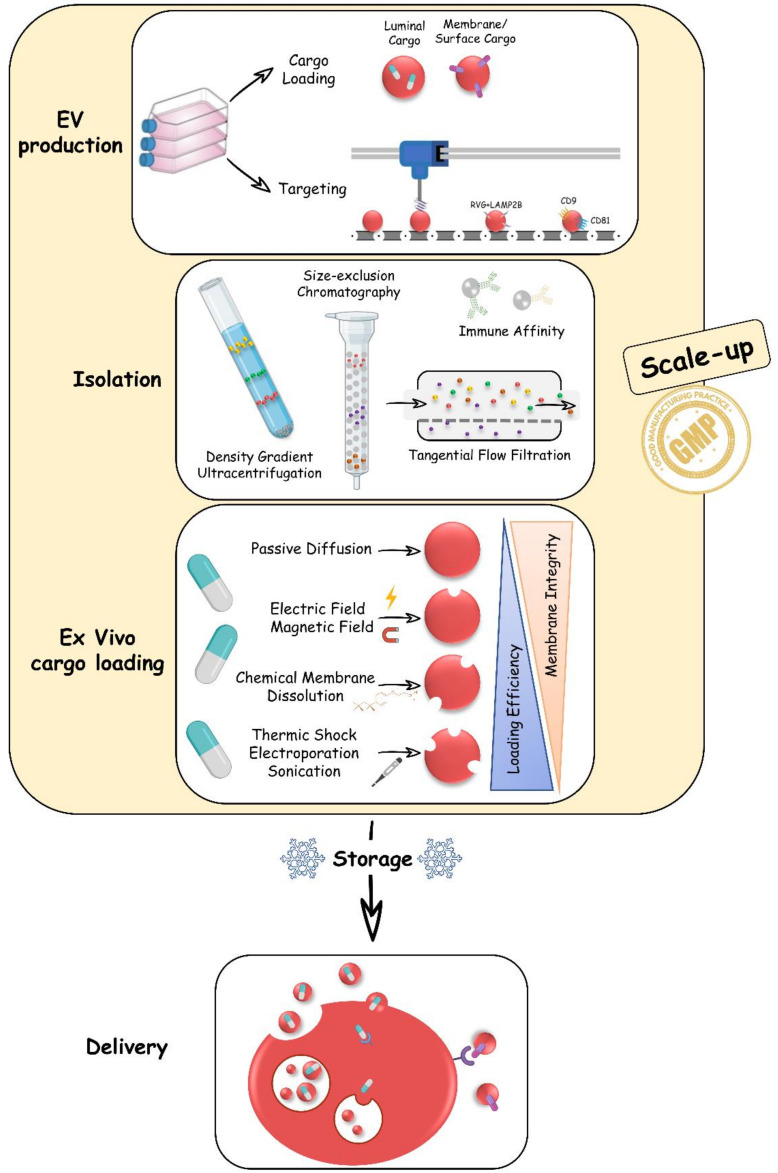
Production of EVs for drug delivery. *Schematic representation of the main passages needed to produce EVs as a drug delivery carrier. Each step must be standardized for the realization of EV-based medicines*.

**Table 1 biology-10-00376-t001:** Therapeutic application of stem cell-derived Extracellular Vesicles (EVs)**.**

Author	Year	Cell Origin	Keywords	Ref.
Martin-Rufino JD.	2019	Mesenchymal Stem Cell	Immunomodulation	26
Balbi C.	2017	Human Amniotic Fluid Stem Cell	Cell proliferation	20
Bian X.	2019	Mesenchymal Stem Cell	Angiogenesis	27
Barile L.	2014	Cardiac progenitor cell	Cardiac regeneration	28
Balbi C.	2019	Human Amniotic Fluid Stem Cell	Cardiac regeneration	29
He J.	2012	Bone marrow stem cells	Renal injury	30
Lo Sicco C.	2017	Mesenchymal Stem Cell	Inflammation	17
Zhang B.	2014	Mesenchymal Stem Cell	Skin burn	31
Fang S.	2016	Umbilical Cord-Derived Mesenchymal Stem Cell	Skin defect	32
Du W.	2017	Mesenchymal Stem Cell	Ischemia	33
Song Y.	2017	Mesenchymal Stem Cell	Sepsis	34
Zhu J.	2018	Mesenchymal Stem Cell	Myocardial infarction	35
Ma J.	2017	Human Umbilical Cord Mesenchymal Stem Cells	Myocardial infarction	36
Feng Y.	2014	Mesenchymal Stem Cell	Myocardial infarction	37
Ibrahim AG-E	2014	Cardiosphere derived cells	Cardiac regeneration	38
Barile L.	2018	Cardiac progenitor cell	Cardioprotection	44

**Table 2 biology-10-00376-t002:** Isolation and purification scale-up and storage cargo loading, EV targeting, and cargo delivery strategies for extracellular vesicle-mediated drug delivery.

	Author	Year	Keywords	Ref
**1. Isolation and purification**	De Jong O.G.	2019	EVs for Drug Delivery	62
	Antimisiaris S.G.	2018	EVs for Drug Delivery	54
	Li P.	2017	EXO Isolation	60
	Cocozza F.	2020	EVs Isolation	63
	Witwer K.W.	2017	MISEV	64
	Karimi N.	2018	EVs Purification	65
	Liangsupree T.	2021	Isolation and Separation	66
	Surman M.	2019	EVs for Drug Delivery	67
**2. Scale-up and storage**	Whitford W.	2019	EXO Manufacturing	68
	Panchalingam K.M.	2015	Scale-up	69
	Andriolo G.	2018	GMP method	21
	Chen Y.S.	2020	GMP EXO	70
	Witwer K.W.	2013	Metodologic standardisation	71
	Lorincz A.M.	2014	Storage	72
	Börger V.	2017	Storage	73
	Bosch S.	2016	Storage	74
	Kaushik J.K.	2003	Storage	75
	Crowe J.H.	1988	Storage	76
**3. Cargo Loading**	Van Eijndhoven M.A.J.	202	RNA drugs	77
	Margolis L.	2019	EVs as API carrier	78
	Katzmann D.J.	2001	Ubiquitinated Proteins and ESCRT	79
	Iavello A.	2016	miRNA and ESCRT	80
	Andreu Z.	2014	Tetraspanin	81
	Frankel E.B.	2018	ESCRT and APIs	82
	Zhang Y.	2010	miRNA overexpression	83
	Pan Q.	2012	RNAi overexpression	84
	Olson S.D.	2012	RNAi overexpression	85
	Cocozza F.	2020	Angiotensin-converting Enzyme 2 overexpression	86
	Zhang Q.	2021	Angiotensin-converting Enzyme 2 overexpression	87
	Sterzenbach U.	2017	Proteins tagging	88
	Liu C.	2019	Proteins tagging	89
	Manfredi F.	2016	Nef exosome-anchoring	90
	Ma J.	2016	Co-incubation of EVs and APIs	91
	Fu S.	2020	Co-incubation of EVs and APIs	92
	Yong T.	2014	Nanoparticles	93
	Illes B	2017	Nanoparticles	94
	Tian Y.	2014	Active cargo loading	95
	Fuhrmann G.	2015	Active cargo loading	96
	Pomatto M.A.C.	2019	Active cargo loading	97
	Kim M.S.	2016	Active cargo loading	98
	Lamichhane T.N.	2021	Active cargo loading	99
	Wang C.	2017	Active cargo loading	100
**4. EV targeting**	Matsumoto A.	2017	Macropahges	101
	Hoshino A.	2015	Tumor	102
	Morelli A.E.	2004	Dendritic cells	103
	Syn N.L.	2017	Cancer therapy	104
	Gomari H.	2018	Cancer therapy	105
	Sugahara K.N.	2010	Cancer drug	107
	Mentkowski K.I.	2019	Cardiac therapy	108
	Hung M.E.	2015	Cancer therapy	109
	Ciullo A.	2019	Cardiac therapy	110
**5. Cargo delivery**	Raposo G.	1996	Ligand-Receptor	3
	Joshi B.S.	2020	Endocytosis	111
	Prada I.	2016	Binding and Fusion	112
	Dimitrov D.S.	2004	Virus entra	113
	Rappa G.	2017	Late endosome	114
	Pedrioli G.	Preprint	Autophagy patway	115

## Data Availability

Not applicable.

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
