# Peer review of "Extracellular Vesicles as Promising Carriers in Drug Delivery: Considerations from a Cell Biologist’s Perspective"

_biology, 2021, doi:10.3390/biology10050376_

Round 1

Reviewer 1 Report

  • The ERP (enhanced permeability and retention) effect and its role in the delivery of exosomes's content (drugs, macromolecules, ect.) should be discussed to cover the exosomes features and nanovesicles tools.
  • The authors missed to mention additional studies about the 3 stages drug delivery systems (DDS) in which exosomes are loaded with synthetic nanoparticles smaller in size (5 - 30 nm) such as carbons nanoparticles (CNPs) or Silica nanoparticles in order to increase the loading efficiency of drugs inside exosomes.
  • It will be helpful for the readers if the authors can add tables summarizing the different studies and clinical trials with their references.

Author Response

The ERP (enhanced permeability and retention) effect and its role in the delivery of exosomes's content (drugs, macromolecules, ect.) should be discussed to cover the exosomes features and nanovesicles tools.

We thank the reviewer for her/his comment. We now added and explained this aspect in the review (line 324-29 page 12).

The authors missed to mention additional studies about the 3 stages drug delivery systems (DDS) in which exosomes are loaded with synthetic nanoparticles smaller in size (5 - 30 nm) such as carbons nanoparticles (CNPs) or Silica nanoparticles in order to increase the loading efficiency of drugs inside exosomes.

We agree with the reviewer's comment. This strategy was not explained in the previous version of the manuscript. In the revised version this was added in the “EV cargo loading” chapter (line 293-96 page 11).

It will be helpful for the readers if the authors can add tables summarizing the different studies and clinical trials with their references.

Two new tables are now present in the revised version of the manuscript. Table 1 summarizes the papers about the therapeutic application of stem cell derived-EVs; Table 2 summarizes all the papers mentioned in the different chapters about the strategies to take into account for Extracellular Vesicles as a drug delivery system.

Reviewer 2 Report

I think the title of this manuscript is too ambitious and does not reflect the main focus given by the authors. The therapeutic potential of extracellular vesicles derived from stem cells.

A cell biology perspective would include the extracellular vesicles from cells affected by different pathologies that have already been shown to contribute to the spread of the disease. An example would be how EVs play an essential role in both primary tumor growth and metastatic evolution. PMID: 32029601 PMID: 32503543 PMID: 31570387 

Concerns:

1-The authors indicate a trial on the effect of modified EVs for the treatment of patients with COVID-19, even giving data, however none of this information can be corroborated due to the lack of references. 

Author Response

I think the title of this manuscript is too ambitious and does not reflect the main focus given by the authors. The therapeutic potential of extracellular vesicles derived from stem cells.

We apologize to the reviewer for the misunderstanding in the title. The main focus of the review was not the therapeutic potential of EVs from a biology perspective but the revision, from a cell biologist point of view, of five aspects that we believe are worth to take into account before considering extracellular vesicles as candidate vectors of pharmaceutical ingredients in cell-free therapies. For such reason, the title of the manuscript is now: “Extracellular Vesicles as Promising Carriers in Drug Delivery: Considerations from a Cell Biologist's Perspective “

A cell biology perspective would include the extracellular vesicles from cells affected by different pathologies that have already been shown to contribute to the spread of the disease. An example would be how EVs play an essential role in both primary tumor growth and metastatic evolution. PMID: 32029601 PMID: 32503543 PMID: 31570387 

We agree with the reviewer comment and we better explain this aspect in the main text in “Therapeutic potential of stem cell derived-EVs” chapter (line 139-44 page 4).

Concerns:

1-The authors indicate a trial on the effect of modified EVs for the treatment of patients with COVID-19, even giving data, however none of this information can be corroborated due to the lack of references

We thank the reviewer for her/his comment. The sentence in the conclusion was now changed (page 14).